# Mobility Based on GPS Trajectory Data and Interviews: A Pilot Study to Understand the Differences between Lower- and Higher-Income Older Adults in Hong Kong

**DOI:** 10.3390/ijerph19095536

**Published:** 2022-05-03

**Authors:** Yingqi Guo, Cheuk-Yui Yeung, Geoff C. H. Chan, Qingsong Chang, Hector W. H. Tsang, Paul S. F. Yip

**Affiliations:** 1Department of Rehabilitation Sciences, The Hong Kong Polytechnic University, 11 Yuk Choi Road, Hung Hom, Hong Kong SAR, China; yingqi.guo@polyu.edu.hk (Y.G.); hector.tsang@polyu.edu.hk (H.W.H.T.); 2Mental Health Research Centre, The Hong Kong Polytechnic University, 11 Yuk Choi Road, Hung Hom, Hong Kong SAR, China; 3Department of Social Work and Social Administration, The University of Hong Kong, Pokfulam, Hong Kong SAR, China; yeungcya@connect.hku.hk; 4Department of Applied Social Sciences, The Hong Kong Polytechnic University, 11 Yuk Choi Road, Hung Hom, Hong Kong SAR, China; geoff.chan@polyu.edu.hk; 5School of Sociology and Anthropology, Xiamen University, Xiamen 361005, China; nkevergreenpine@163.com; 6Hong Kong Jockey Club Centre for Suicide Research and Prevention, The University of Hong Kong, 5 Sassoon Road, Pokfulam, Hong Kong SAR, China

**Keywords:** Global Positioning System (GPS), mobility, activity space, social exclusion, older adults, Hong Kong

## Abstract

Few studies have examined mobility from a social exclusion perspective. Limited mobility can restrict opportunities to interact with others and therefore may lead to social exclusion. This pilot study was designed to test the feasibility of integrating Global Positioning System (GPS) trajectory data and interview data to understand the different mobility patterns between lower- and higher-income older adults in Hong Kong and the potential reasons for and impacts of these differences. Lower- (n = 21) and higher- (n = 24) income adults aged 60 years of age or older in Hong Kong were recruited based on purposive sampling. They were asked to wear a GPS device for 7 days. Seven measures of mobility (four dimensions) were created based on GPS data and compared between lower- and higher-income older adults, including extensity (standard deviation ellipse, standard distance between all locations), intensity (time spent out of home, doing activities), diversity (number of locations), and non-exclusivity (time spent in public open spaces and places with higher public service provisions). It then administered semi-structured interviews to understand the determined differences. The activity spaces for lower-income older adults were, on average, smaller than those for higher-income older adults, but lower-income older adults spent significantly more time participating in out-of-home activities. They were more likely to be exposed to environments with similar socioeconomic characteristics as their own. The interviews showed that limited social networks and expenditure on transport were the two main factors associated with lower-income older adults having relatively fewer activity spaces, which may lead to further social exclusion. We recommend using GPS in daily life as a feasible way to capture the mobility patterns and using interviews to deeply understand the different patterns between lower- and higher-income older adults. Policy strategies aiming to improve the mobility of older might be helpful for further improving the social inclusion of lower-income older adults.

## 1. Introduction

Worldwide, the percentage of older people in the population is growing faster than any other age group. According to World Population Prospects [1], one in six people will be aged over 65 years old by 2050 (16%), this figure being almost double the 2019 estimates of one in eleven (9%). People worldwide are living longer due to improved living circumstances, education, and health care. However, living a long life does not necessarily equate to living a quality life; hence, maintaining the quality of life for older people is an important public health concern [2]. Quality of life is multifactorial and includes health and social functioning [2]. The World Health Organization (WHO) promotes the concept of healthy aging, suggesting that mobility plays a pivotal role in maintaining older adults’ health, independence, and participation in society [3]. Mobility maintenance is fundamental to healthy aging [4]. Large population-based studies examining aspects of healthy aging have positively linked mobility to health status and quality of life [4,5,6,7,8]. Few studies have explored mobility from a social exclusion perspective.

Social exclusion not only involves a lack of goods, services, resources, and rights, but also limited activity spaces, an inability to participate in normal activities, and a lack of access to activities that should be able to be enjoyed by all [9]. Suttles’ (1972) seminal work on the social structure of communities provides a foundation for research that explores how social and symbolic boundaries shape the spatial patterns of different groups [10]. As Bourdieu (1999) indicates, a person’s social position is expressed by the site of the physical space within which that person exists. Location is important in social relationships, as people move from one space to another [11]. Empirical studies have explored different residential and mobility patterns between people with different socioeconomic status (SES). For example, Xu [12] compared the mobility patterns of different SES groups among the general populations in Singapore and Boston. They found that richer people travelled shorter distances in Singapore, but not in Boston, which may indicate differences in city structures and ease of transport. Garas [13] suggested that mobility varied by SES because of different choices in modes of transportation. Pappalardo [14] found that mobility at the municipal level in France was positively associated with income and educational levels and negatively associated with deprivation among the general population. Yip [15] suggested that people aged over 18 years old from richer neighborhoods in Hong Kong travelled into other rich neighborhoods more frequently than poorer people did. In another Hong Kong study, Wang [16] found that residents in low-income public housing had more extensive activity spaces and spent more time outside their homes compared to those living in private housing. Whilst all these studies explored mobility from a social exclusion perspective, most were conducted among general populations. Little is known about whether mobility inequality and subsequent social exclusion occurs in older people. Nutley [17] noted that older adults had different mobility patterns compared to younger adults, such as shorter travel distances and a higher frequency of moving around a permanent home for health and leisure needs. Therefore, it is reasonable to hypothesize that the differences in mobility patterns reported between lower and higher SES groups in younger ages might be less noticeable in older adults.

Hong Kong is a typical high-density Asian city that is undergoing rapid population aging. However, unlike many other cities in high income Western countries, Hong Kong’s high population density supports an efficient and affordable transportation system, where most residents live in relative proximity to public services. In 2019, private cars were used by only 76.3 per 1000 people in Hong Kong, with most people using public transport [18]. Franchised buses were the most common form of transport followed by mass transit railways and public light buses [19]. Most trips were within 30 min of home and were mainly to visit markets and food places [19]. Transportation and public services in Hong Kong are well provided, widely accessible, and convenient in both lower- and higher-income areas; therefore, it is reasonable to hypothesize that findings from Western studies on a mobility gap based on SES may not hold true for Hong Kong. Furthermore, adults aged 65 years old or over only have to pay HKD 2.00 for each trip, which is a strategy of the Hong Kong Government to promote the mobility of older adults.

Activity spaces, which are defined as the areas within which people travel in their daily life, are an important concept and are used as a proxy for a person’s mobility. Traditionally, the most commonly used method to collect activity space data is a time-space technique from the 1960s based on personal diaries [20]. Participants retrospectively recorded their activities over a period of time (ranging from the previous week or the previous day). This method has apparent disadvantages. For example, these diaries are based on recall and cognition [21], which may lead to inaccuracy or loss of important information. With the development of tracking technologies such as the Global Positioning System (GPS), obtaining accurate information on an individual activity spaces can now be realized. The development of GPS devices has created the possibility of gathering real-time spatial and temporal data [22,23]. Over the past few decades, the cost and size of GPS devices have decreased, and function has increased, enabling research into individual mobility and activity spaces [24,25]. Compared to traditional static design, which applied questionnaire surveys to collect information regarding resident mobility and activity spaces, the in situ study design can be conducted in the daily lives of residents and plays a pivotal role in reducing recall bias. Therefore, it is worth testing whether applying these wearable technologies can successfully capture the real-time daily mobility and activity space information without asking the participants to complete a survey questionnaire that requires them to recall their activities from past days. Moreover, to the best of our knowledge, no research has been conducted using qualitative data (interview) to understand the underlying mechanism between the different patterns captured by quantitative real-time data collected by wearable devices. This paper adds to the scarce body of literature about the real-time and dynamic mobility patterns and potential reasons for the different patterns between older Hong Kong adults in high and low SES groups.

The study aims were to:(1)Test the feasibility of using GPS tracking data to quantify older adults’ mobility patterns and exposure to social environments and integrate interview data to understand the potential reasons;(2)Compare the mobility and social environmental exposure patterns between lower- and higher-income older adults based on pilot GPS data;(3)Explore the potential reasons for different patterns between lower- and higher-income older adults in Hong Kong based on the interview data.

## 2. Materials and Methods

### 2.1. Design

This pilot study aimed to test recruitment processes, establish the feasibility of extracting mobility indicators from GPS data, and identify underlying reasons for differences in mobility patterns between lower- and higher- income older adults based on semi-structured interviews. This study was approved by the Human Research Ethics Committee for Non-clinical Faculties, the University of Hong Kong (Ref: EA1508040).

### 2.2. Participants

Healthy adults aged 60 years old and older, living independently, and who could walk without assistance were recruited from two districts in Hong Kong with a mixed SES status (North District and Tseung Kwan O). The contexts of these districts are generalizable to other densely populated areas in Hong Kong such as the New Territories and Kowloon districts. Lower income was defined as receiving government comprehensive social security assistance (CSSA)), which is not available to individuals with higher incomes.

We recruited 45 older adults from two pilot districts via purposive sampling (i.e., also known as judgmental, selective, or subjective sampling) using “CSSA (i.e., lower-income) or Non-CSSA (i.e., higher income)” as our main characteristic, with each group having at least 20 participants with diverse characteristics (i.e., age, gender, martial status, educational level, health status). Potentially relevant older adults were identified by the trained senior research assistant with help from our project collaborators in these two pilot districts. Potentially relevant people were invited to participate in the research by the senior research assistant on the phone. The participants signed a consent form and agreed that they would wear a GPS device at all times for one week and attend a short semi-structured interview (up to 30 min). The participants could only remove the GPS when sleeping or bathing. They could choose to stop participating in the research at any time without negative consequences. Those who completed the study requirements received HKD 200 in appreciation. The number of people required for the sample and the duration of the study period were decided following previous studies, i.e., a literature review reported that most studies applying GPS to track activity spaces only involved a few people (less than 40 people) over a short period of time (less than 12 days) [26]. The third author had expertise in psychological statistics and guided the sampling and data collection procedures.

### 2.3. GPS Mobility

The GPS devices (A9 brand) were purchased online. They could be placed in a pocket, a bag, or on other places on the body as required.

Mobility was measured using a comprehensive framework for studying the social implications of activity spaces by Wang [16]. The framework used four characteristics of activity spaces to assess an individual’s daily activity patterns related to social inclusion or exclusion: (1) Extensity: the spatial dispersal of activity spaces to measure the range and the ability to reach opportunities; (2) Intensity: the frequency and duration of visits to certain places to measure the significance of these places in individual’s daily life; (3) Diversity: the number of different locations in one’s activity space to measure the richness of one’s daily life; (4) Exclusivity (/non-exclusivity): the degree of isolation of one’s activity space involving the use of private versus public open spaces.

The geographical extensity of the individuals’ daily lives was assessed by the standard deviation ellipse (SDE) and standard distance between all of the locations in the activity space. The SDE delineates geographical distribution trends by summarizing both the dispersion and orientation of the observed samples. It calculates the standard deviation of the x-coordinates and y-coordinates from the mean center to define the axes of the ellipse. The standard distance measures the compactness of a distribution with a single value the represents the dispersion of features around the center. The value is distance; therefore, the compactness of a set of features can be represented on a map by a circle with the radius equal to the standard distance value. Figure 1 illustrated examples of SDE and standard distance. The intensity of the out-of-home activity space was measured by the time people spent outside of their homes and the time spent in out-of-home activity locations (i.e., excluding travel time). The activity location refers to the places where the participants spent more than 15 min within 150 m of. The number of out-of-home activity locations measured the diversity of an individual’s activity space. The non-exclusivity of one’s activity space was assessed through the use of urban public spaces, including the percentage of one’s activity time spent in public open spaces, and the time spent in places with higher than average public service provisions (i.e., Large Street Block (LSB) as a unit of measurement). ArcGIS version 10.2 (Environmental Systems Research Institute (ESRI), Redlands, California, U.S.) was used to calculate the above indicators.

Following the four-dimensional framework, we compared the activity spaces of lower- and higher-income older adults’ using the seven measures shown in Table 1.

An important element of social isolation is the lack of interaction with different social groups. Social environmental exposure therefore become a popular measure of social isolation that assess the social homogeneity in a space. The underlying assumption is that the chance of group interaction depends on the presence of different groups in one’s activity space. The higher the proportion of different SES groups, the more likely it is that a person will interact with people different from his/her own group, and there, he/she would be then less socially isolated. One’s activity space would therefore be further characterized by the SES composition of the people who are present in one’s activity space. Following Wang’s study [16], we applied 2016 census data, which include information on the population composition of small areas, to calculate the social environmental exposures of an individuals’ activity space. Variables used to describe the SES composition of an activity space include the median income of household, the percentage of less educated people (i.e., primary or below), the percentage of people with non-professional jobs (i.e., cleaners, transport drivers), the percentage of single-parent families, and the percentage of public housing households. These census-based variables were weighted by the amount of time an individual spent in the related census area. It was assumed that more exposure to higher SES groups may contribute to lower-income residents having greater social integration, while more exposure to people in similar lower SES groups as themselves may indicate greater levels of social isolation for lower-income residents, and vice versa.

### 2.4. Semi-Structured Interview

The senior research assistant conducted one semi-structured interview that was less than 30 min long with each participant. Interview questions included: (1) Where did you go in the last week? (2) What did you do in these that places you went to in the last week? (3) Whom did you meet with in last one week? (4) Did you met any barriers in going to the places where you wanted to go? (5) Are you satisfied with your community environment? (6) What aspects do you think can be improved regarding your community environment? (7) How is your financial status and health status? The semi-structured nature of the interview enabled the researcher to probe more deeply should answers to the interview questions provide opportunities to learn more. Together with the senior research assistant, one research assistant attended the interview as a dependent observer. The interviews were recorded.

### 2.5. Analytical Method

The GPS data, which included a table consisting of time and position, were exported from the devices and imported into ArcGIS version 10.2. Seven activity space indicators as well as the social environmental exposure were calculated based on the time-position table. We compared the differences in the activity space indicators and social environmental exposure between the lower- and higher-income older adults using ANOVA analysis. Regarding the qualitative interview data, a thematic analysis was conducted. The recordings were transcribed verbatim, translated from Cantonese to English, and cross-checked by two research assistants. The first author coded and analyzed all of the interview transcriptions following an applied health social science lens through an in-depth reading by using Nvivo Version 12 [27]. As we aim to identify the potential themes that can be automatically determined using the data rather than by considering expected themes based on existing theory, inductive thematic analysis was applied. A stage-by-stage procedure was used to identify emerging themes, which were further grouped and combined to obtain more coherent and relevant themes. We paid attention to those frequently recurring expressions and recurring ideas comprising the major themes. The main aim of the inductive thematic analysis was to provide a clearer understanding of the underlying potential reasons for and implications of any differences in the mobility patterns identified between lower- and higher- income older adults. This information also helped us to understand the unique situation of older people in Hong Kong and to inform public policies on social inclusion.

## 3. Results

All 45 older adults approached to participate in the study consented to participate and completed the study. The participant characteristics are summarized in Table 2.

Figure 2 illustrates where the participants lived and the extensity patterns based on SDE. Table 3 reports differences in the activity spaces between lower- and higher-income older adults using Wang’s seven variables. As hypothesized, there were no differences between the two groups in terms of activity space extensity and diversity. However, the extensity of the activity space was, on average, higher among higher-income older adults than lower-income older adults. Moreover, despite a smaller activity space, lower-income older adults spent longer amounts of time participating in out-of-home activities. This suggests that lower-income older adults may spend more time outside of their homes at specific places for various activities. Lower-income older adults also spent more time at places with a higher provision of public services than higher-income older adults. This suggests that spaces with higher levels of public service provisions are probably a more important part of the daily lives of lower-income older adults. By contrast, higher-income older adults might be more likely to go further away to where there may be higher quality and more choice in services.

Table 4 compares the social environmental exposure of lower- and higher-income older adults to experiences in their daily lives. The results show that the two groups demonstrate significant differences in terms of all five exposure variables. Lower-income older adults spent more time in environments that had similar levels of social class as themselves, including neighborhoods with lower median household income, groups of people with lower levels of education, non-professional jobs, single-parent families, and public housing households.

Table 5 reports the three dominant themes related to the GPS dimensions as well as the example quotations. The information extracted from the interview transcriptions provide a deeper and broader understanding about the above findings identified from the GPS data. Some lower-income older adults reported that many reasons reduced their mobility to areas that were further away. The three main themes that were identified based on qualitative interview were summarized as follows: (1) social network (i.e., seven participants mentioned that their social circle was small and that they did not often have contact with friends in lower-income groups, while two cases reported that they visited friends in a higher-income group often); (2) transport costs (i.e., four cases mentioned that transport expenditure was a barrier that prevented them from going to places where they wanted to go, while no one mentioned transport expenditure as an issue in the higher-income group); (3) public services (i.e., three lower-income cases mentioned that the public services around their living place met their daily needs, while in the higher-income group, two cases shared that they rarely used the public services near their home and that they used paid facilities further away).

## 4. Discussion

Studies from other countries have shown that older adults with lower incomes are disadvantaged in terms of social inclusivity, as their choices can be constrained by lower educational levels, a higher prevalence of chronic disease, less frequent interaction with friends, and constrained mobility [28,29,30]. We hypothesized that due to Hong Kong’s well-developed, affordable transport system and wide availability of health and social services, income would make no difference to the daily activity patterns of older adults. However, we found inequality between lower- and higher-income older adults in terms of activity space and environmental exposure.

The average activity spaces of lower-income older adults were generally smaller than those of higher-income older adults, although the difference was not significant. This finding might be due to the relatively small sample size or lack of variability in the sample. One reason for the generally smaller activity space of lower-income older adults might be because they have a more routine lifestyle than higher-income older adults who are of a similar age and health status. The lower-income participants may have had a smaller social network and hence less contact with others compared to higher-income persons [31]. This may indicate a lack of social support, which may further lead to social isolation. This can prevent lower-income people from learning about and accessing resources to lift them from poverty [32] and may further lead to a vicious cycle that entrenches and aggravates poverty [33]. This indicative finding requires testing in larger more diverse samples.

Contrary to our expectations, our study found that lower-income older adults spent longer times in out-of-home activities than wealthier older people. This might be explained by the small living spaces in Hong Kong, where half of the homes are less than 500 square feet in size [34]. Lower-income older adults living in small homes may prefer to go out to conduct their daily activities instead of staying at home. Comparatively speaking, wealthier people’s homes may be more comfortable and have more space and greater be more convenient for activities such as watching TV, preparing and eating meals, reading books, and doing basic exercises such as tai chi.

The transport subsidy for older adults in Hong Kong (flat rate of HKD 2.00 (USD 0.25) for a trip) makes the regular use of public transport affordable, potentially improving the access of older adults to out-of-home activities. We found that lower-income older adults spent more time in areas with a higher provision of public services such as public parks, libraries, and community centers compared to higher-income older adults. Hong Kong Government planning has ensured that there is no difference in the public service provisions in Hong Kong between lower- and higher-income areas [35]. This means that lower-income older adults can access public open and free spaces just as easily as richer people can. The provision of services is related to the size of the population in a district; for example, public libraries and swimming pools are available for districts with more than 50,000, and 100,000 residents, respectively. However, richer people may have greater access to private services such as swimming pool and dancing studios, while lower-income older adults may spend more time in public spaces that are more affordable or free, such as parks and public open spaces.

Our findings suggest that lower-income older adults were more likely to access environments with similar SES characteristics as their own. A lack of exposure to higher levels of social class groups may constrain their opportunities to engage in cross-group interactions and support [16]. The separation between lower- and higher-income groups may be extended due to differential patterns of activity space use [36]. Although some studies suggest that social exclusion in residential places is more significant than non-residential areas, an examination of the differences in exposure between lower- and higher-income residents in their non-residential areas leads to a better understanding of the barriers to social interaction across different social classes [37].

To the best of the authors’ knowledge, this pilot study leads the way in investigating mobility and social environmental exposure in older adults in an Asian context by using in situ GPS tracking to collect real-time location data and activity space data during daily life. None of the participants found the device difficult to wear, and none withdrew from the study. The objective indicators that were extracted from the GPS data were useful to measure mobility, and there was value in using additional interview data to explore income differences in social environmental exposure.

The pilot study identified issues that will improve the current design and conduct of future larger scale studies. First, more in depth information is required from interviews. For example, the reasons why lower-income older participants were more likely to go to areas with a similar SES as themselves remain unclear. It could reflect a lack of resources, a lack of knowledge, or a constrained social network, but it may also reflect a choice that has little to do with income. More in-depth questions need to be designed in future studies to reveal the deeper reason for this pattern. Second, larger representative samples based on probability sampling methods (i.e., simple random sampling, systematic random sampling, stratified sampling, and cluster sampling) are required to examine the relationships between income and mobility that can be generalized across Hong Kong. Based on the purposive sampling in this pilot study, the recruited older adults were more likely to be female and healthier compared to the population of older adults in Hong Kong. It is reasonable to hypothesize that the mobility gap between lower- and high-income older adults with chronic diseases might be more apparent. Future studies including more representative samples may be able to capture a more holistic picture of Hong Kong. Third, the study design should incorporate investigation of factors that potentially mediate the link between poverty and mobility, such as social capital, social network, and social support. More information about the participants’ daily lives could be obtained if an ecological momentary assessment (EMA) study design was used. Fourth, new developments in advanced remote sensing techniques, mobile phones, and applications can open the way to investigating people’s behaviour in entire urban regions without the need for people to wear a GPS device [38]. These techniques could be helpful in collecting more large information for mobility research. Fifth, in this pilot study, we also found that the battery in the GPS device was sometimes unable to maintain its charge for the entire research period, and it is unknown on which day the device may have run out of battery. Therefore, we required our participants to charge the device frequently, which represents a burden placed on the participants. Furthermore, matured devices that can last longer will be explored in future studies.

## 5. Conclusions

This pilot study suggests that it is feasible to use GPS trajectory data to quantify mobility and exposure to social environments and used interview data to explore the underlying reasons for different patterns between lower- and higher-income older adults in Hong Kong. This pilot study also identified issues that will need to be improved in future studies, i.e., including larger representative samples with random sampling methods rather than purposive/convenient/simple sampling methods. The preliminary findings of this pilot study suggest that there may be mobility inequalities between lower- and higher-income older adults in Hong Kong despite its high population density and well-developed transportation system. This needs to be confirmed in larger population studies. If these inequalities do exist, strategies aimed at providing more support for lower-income individuals to expand their mobility range and improve the diversity of the places they visit might enhance their social inclusion. For example, the Rehabilitation Programme Plan (RPP) 2020 published by the Rehabilitation Advisory Committee (RAC) in Hong Kong suggests directions for traffic improvements, such as service efficiency, service capacity, and adopting Information and Communication Technology (ICT). However, two years have already passed since then, and the implementation progress has not been satisfactory, something that is mainly due to the COVID-19 pandemic. Using an ICT framework to implement smart transport programs might be a helpful way to facilitate access to more convenient transportation. Moreover, in 2022, the Hong Kong government has started to extend the HKD 2.00 support to more forms of transport (i.e., Red Minibus Route and Kaito Route), which will provide more opportunities for older people from different SES backgrounds to congregate. In addition, a transport policy study conducted in Hong Kong suggests that shortening walking and waiting times can improve the probability of older adults making a trip, and improving seat availability can be an effective way to achieve this [39]. People living in Hong Kong, especially older adults, are deprived of private space. The availability and accessibility to public space acts as a form of compensation for this deprivation. During the COVID-19 pandemic, almost all public facilities closed due to quarantine measures, and the disruption of these services has had a potentially devastating impact on individuals and their communities, having a particular effect on older adults [40]. Understanding the different mobility patterns and preferences of the mode of movement between different SES groups will inform strategic planning directions for a more livable and socially inclusive city for older adults in similar high-density countries.

## Figures and Tables

**Figure 1 ijerph-19-05536-f001:**
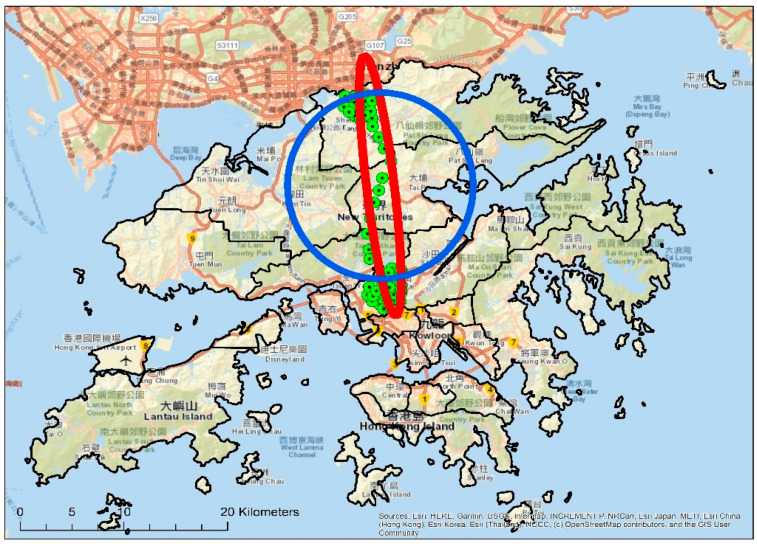
Example of SDE (red) and standard distance (blue).

**Figure 2 ijerph-19-05536-f002:**
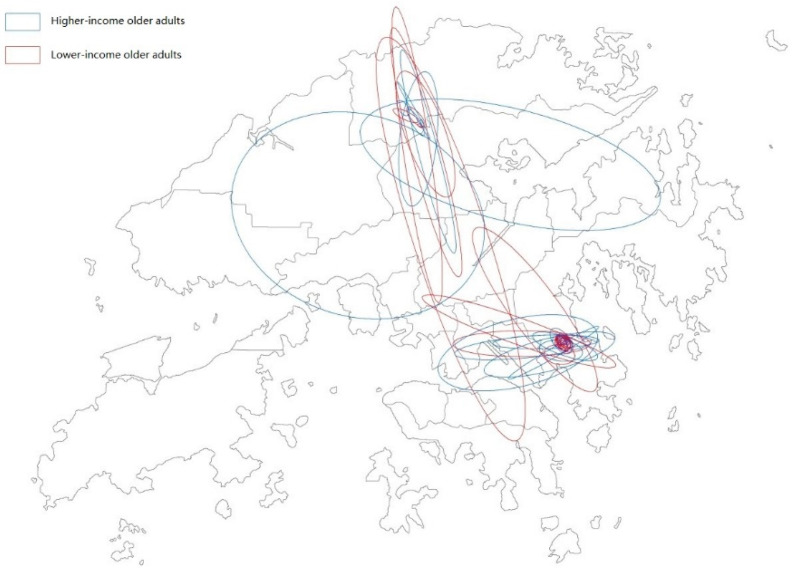
Participants’ residential places and extensity pattern based on SDE.

**Table 1 ijerph-19-05536-t001:** Variables measuring the four characteristics of activity space.

Dimensions	Variables
Extensity	Standard deviational ellipse (SDE)
	Standard distance to daily activity locations
Intensity	Total time spent out-of-home
	Total time spent on out-of-home activities (excluding travel time)
Diversity	Number of out-of-home activity locations visited
Non-exclusivity	% of activity time spent at public open spaces
	% of activity time spent at places with averagely higher public service provision

**Table 2 ijerph-19-05536-t002:** Sample profile.

Variable		Lower-Income Group (n = 21) Mean (S.D.)/N (Percentage)	Higher-Income Group (n = 24) Mean (S.D.)/N (Percentage)
Age	Range 60–87	70.8 (8.2)	68.9 (7.4)
Gender	Female	16 (76.2%)	17 (70.8%)
Marital status	Married	11 (52.4%)	12 (50.0%)
Educational level	Primary and below	12 (57.1%)	11 (45.8%)
Healthy status	Have chronic disease	6 (28.6%)	11 (45.8%)

**Table 3 ijerph-19-05536-t003:** Comparison of activity spaces between lower- and higher-income older adults.

Dimensions	Variables	Lower-Income Group	Higher-Income Group	Total
Extensity	Standard deviational ellipse (SDE)	7.6	9.5	8.6
	Standard distance between daily activity locations	9.8	11.6	11.1
Intensity	Total time spent out-of-home	8.8	8.5	8.7
	Total time spent on out-of-home activities	6.9	4.2 *	5.3
Diversity	Number of out-of-home locations visited	2.3	2.4	2.3
Non-exclusivity	% of activity time spent at public spaces	88.4	90.2	89.5
	% of time spent at places with higher public service provision	69.7	65.3 *	68.2

* *p* < 0.1.

**Table 4 ijerph-19-05536-t004:** Comparison of exposure index between lower- and higher-income older adults.

Variables	Lower-Income Group	Higher-Income Group
Median income of household	25,000	27,000 *
The percentage of lower-educated people	69.4	62.9 *
The percentage of persons with non-professional jobs	54.2	50.8 *
The percentage of single-parent families	13.1	10.1 *
The percentage of public housing households	42.2	27.9 **

* *p* < 0.1; ** *p* < 0.05.

**Table 5 ijerph-19-05536-t005:** Dominant themes related to GPS dimensions and example quotations from interviews.

Themes	GPS Dimensions	Example Quotations
Social network	Extensity, Diversity	“I didn’t meet anyone except my wife in last week…I went to these places by myself” (007, lower-income 65 years male); “I didn’t contact with friends and children for long time…” (010, lower-income 61 years female) “I went there by myself…no one accompanied me…” (023, lower-income 67 years male) “My friends live far from here… we haven’t contacted for long time…” (025, lower-income 74 years male) “My friends were busy in caring grandson…we didn’t meet frequently…” (026, lower-income 72 years female) “I went there by myself, no one accompanied me…I didn’t visit any one in last one week” (033, lower-income 70 years female) “They don’t live here… we didn’t meet much…long time…yes haven’t met…” (037, lower-income 69 years female) “By myself…my friends were busy in caring grandchildren…we haven’t met for long time…” (040, lower-income 72 years male) “I didn’t visit anyone in last one week” (042, lower-income 70 years female) “I went hiking four times in last week with my friends…” (032, higher-income 69 years female) “I went shopping in Causeway bay with my two friends… We together every week…” (029, higher-income 72 years female)
Transport cost	Extensity	“My best friend lives in Sai Ying Pun, however it cost around…10 more…yes around 12 dollars from my home to go… ” (003, lower-income 63 years female) “I didn’t go other areas in last 7 days I just stay in community…if no need pay transport I would go” (010, lower-income 61 years female) “My income is… my daughter gave me…sometimes… some months…it cost much going to Hong Kong Island…I got a part-time olders’ work there but expenditure cost… ” (023, lower-income 67 years male) “Older adults’ transport fee is still high…” (041, lower-income 65 years female)
Public services	Intensity, Non-exclusivity	“I’m satisfied with community facilities, I do exercises in rest park…” (009, lower-income 66 years male) “I didn’t go other districts in last 7 days, I can do all things around home…” (031, lower-income 72 years female) “Service enough for me…no need cost…” (042 lower-income 70 years female) “I went to Kwai Qing for massage…the service is famous and good there…” (019 higher-income 70 years male) “The tennis table crowded sometimes… I sometimes went to sport center in Hong Kong Island…meanwhile visit my daughter…” (044 higher-income 72 male)

## Data Availability

The summary of GPS trajectory data can be accessed via contacting the corresponding author. The geographical and socioeconomic data were obtained by application from a third party (the Hong Kong government). The authors had no special access privileges to these data. Interested researchers can obtain the data by request or purchase from the Hong Kong Government by contacting relevant Departments—Census and Statistics Department for the census data and Lands Department for the Geo-Community Database and the Geo-Reference Database. Instructions for data access can be obtained at the websites of the respective agencies; please find websites and contact information below—Census and Statistics Department: (http://www.censtatd.gov.hk/home/index.jsp, accessed on 28 April 2022); Land Department: (http://www.landsd.gov.hk/en/about/welcome.htm, accessed on 28 April 2022).

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
