# Peer review of "Mobility Based on GPS Trajectory Data and Interviews: A Pilot Study to Understand the Differences between Lower- and Higher-Income Older Adults in Hong Kong"

_ijerph, 2022, doi:10.3390/ijerph19095536_

Round 1
Reviewer 1 Report
Dear authors,
Many thanks for properly addressing all my comments. The paper has been significantly improved.
Best regards,
Author Response
- “Many thanks for properly addressing all my comments. The paper has been significantly improved.”
Response:
Thanks!
Reviewer 2 Report
I am satisfied with the corrections made by the authors in the revised version. The paper can be accepted for publication. Thanks. Only some minor revisions: Figure 1 resolution better be improved a bit. After Conclusion before "...can shed light" some words are missing, incomplete sentence. Please address.
Author Response
- “I am satisfied with the corrections made by the authors in the revised version. The paper can be accepted for publication. Thanks. Only some minor revisions: Figure 1 resolution better be improved a bit. After Conclusion before "...can shed light" some words are missing, incomplete sentence. Please address. ”
Response:
Thanks! We have improved the resolution of Figure 1, and revised conclusion part accordingly.
Reviewer 3 Report
It seems that the authors have tried their best to revise the manuscript. But they have not addressed my concerns. They turned the research design into “the comparative and mixed-design” to relax the limitations of less sufficient and representative samples. But it still didn’t change the fact that the sample is not representative. In addition, there is no detailed description of the step and result of inductive thematic analysis, which may raise questions about whether the method has been properly applied. In my opinion, the issue-related research design is not rigorous and the novelty is marginal, so this paper should not be accepted for publication.
Author Response
- “It seems that the authors have tried their best to revise the manuscript. But they have not addressed my concerns. They turned the research design into “the comparative and mixed-design” to relax the limitations of less sufficient and representative samples. But it still didn’t change the fact that the sample is not representative. In addition, there is no detailed description of the step and result of inductive thematic analysis, which may raise questions about whether the method has been properly applied. In my opinion, the issue-related research design is not rigorous and the novelty is marginal, so this paper should not be accepted for publication.”
Response:
Thanks for the comments. Indeed a larger representative samples based on probability sampling methods can better represent all Hong Kong older adults and can be more readily generalized to Hong Kong situation, compared with the non-probability sampling methods. We have stated clearly in the limitation section, it is a kind of pilot study to make use of the first of this kind study to examine the mobility between the poor and non-poor older adults. In addition, we have discussed the potential bias might be caused based on the present study which based on purposive sampling, as follows:
“Second, larger representative samples based on probability sampling methods (i.e., simple random sampling, systematic random sampling, stratified sampling, and cluster sampling) are required to examine relationships between income and mobility, that can be generalized across Hong Kong. The recruited older adults based on purposive sampling in this pilot study are more likely to be female and those healthier compared to Hong Kong general older population. It is reasonable to hypothesize that the mobility gap between lower- and high-income older adults with more chronic disease might be more apparent. Future studies including more representative samples can capture a more holistic picture in Hong Kong.”
Regarding the purposive sampling procedure, we have added more information about how we recruited the participants, emphasized the “CASS (i.e., lower-income) or non-CSSA (i.e., higher-income)” as the main purposive, and included the reference we used to guide our sampling procedure.
“We recruited 45 older adults from two pilot districts via purposive sampling (i.e., also known as judgmental, selective, or subjective sampling) by using “CSSA (i.e., lower-income) or Non-CSSA (i.e., higher income)” as our main purposive characteristic, with each group having at least 20 participants with diverse characteristics (i.e., age, gender, marital status, educational level, health status)…… The number of sample size and study period were decided following previous studies, i.e., a literature review reported that most studies applying GPS to track activity spaces involved few persons (less than 40 persons) over short time periods (less than 12 days) [26]. The third author expertized in psychological statistics guided the sampling and data collection procedure.”
Actually, this study was originally designed to pilot the feasibility of integrating GPS mobility data and interview data to compare the different mobility pattern between lower- and high-income older adults and to understand the potential underlying reasons. This is our first step to move from questionnaire survey data (static design) to GPS mobility data (dynamic design) among older adults (i.e., reduce the re-call bias), we have added more about why we are trying to adopting a dynamic design in introduction and we have summarized the main problems we met in our pilot in discussion. Throughout the paper, we have emphasized more about this the feature of this pilot trying.
“Compared to traditional static design which applied questionnaire survey to collected residents’ mobility and activity space, the in-situ study design conducted in residents’ daily life plays pivotal roles in reducing the recall bias. Therefore, it is worth to test whether applying these wearable technologies can successfully capture the real-time daily mobility and activity space without asking the participants to conduct survey questionnaire which needs them to recall back their memory in past days. Moreover, to the best knowledge, no research has been conducted to use qualitative data (interview) to understand the underlying mechanism between the different patterns captured by quantitative real-time data collected by wearable devices. This paper adds to the scarce body of literature about the real-time and dynamic mobility pattern and potential reasons for the different patterns between older Hong Kong adults in high and low SES groups.”
“The pilot study identified issues which will improve the current design and conduct of future larger scale studies. First, more in depth information is required from interviews. For example, the reasons that lower-income older participants were more likely to go to areas with similar SES as themselves remain unclear. It could reflect lack of resources, lack of knowledge or a constrained social network, but it may also reflect choice that has little to do with income. More in-depth questions need to be designed in future studies to reveal the deeper reason. Second, larger representative samples based on probability sampling methods (i.e., simple random sampling, systematic random sampling, stratified sampling, and cluster sampling) are required to examine relationships between income and mobility, that can be generalized across Hong Kong. The recruited older adults based on purposive sampling in this pilot study are more likely to be female and those healthier compared to Hong Kong general older population. It is reasonable to hypothesize that the mobility gap between lower- and high-income older adults with more chronic disease might be more apparent. Future studies including more representative samples can capture a more holistic picture in Hong Kong. Third, the study design should incorporate investigation of factors potentially-mediating the link between poverty and mobility, such as social capital, social network, and social support. More information about participants’ daily lives could be obtained if an ecological momentary assessment (EMA) study design was used. Fourth, new developments of advanced remote sensing techniques, mobile phone and application can open the way to investigating people’s behaviour in entire urban regions without the need for people to wear a GPS device [38]. These techniques could be helpful in collecting more large information for mobility research. Fifth, in this pilot study, we also found that the battery of GPS sometimes cannot support for whole research period, the research doesn’t know which day which device may run out of battery. Therefore, we required our participants charged frequently which might cause burden to participants. Furthermore, matured devices which can last longer time will be explored in future study. ”
Regarding the qualitative part, we have improved its volume to be similar as the GPS part by adding more details about each step (i.e., data collection, data analysis, and results interpretation) throughout the paper. In addition, we have added it in the title by emphasizing its similar important role compared to GPS part.
We agree very much that a larger sample can better be generalized to Hong Kong older adults. Based on the trying of this pilot, we will adjust some details in the data collection procedure and recruit more participants to further conduct a larger probability sampled study as mentioned in discussion.
Reviewer 4 Report
Lines 103-117: please elaborate research gap (lines 103-105: one sentence without any references might not be sufficient to explain why this research value) and contributions clearly.
There are two conclusions. which one is the one to review? also Lines 368-370: this is not complete sentence,
It would be helpful to add this reference since this study was about HongKong as well.
Public transport policy measures for improving elderly mobility
https://doi.org/10.1016/j.tranpol.2017.12.015
Author Response
- “Lines 103-117: please elaborate research gap (lines 103-105: one sentence without any references might not be sufficient to explain why this research value) and contributions clearly. ”
Response:
We have revised the research gap, as highlighted from line 110-120. Thank you very much for the very helpful suggestion which can enhance the paper a lot!
- “There are two conclusions. which one is the one to review? also Lines 368-370: this is not complete sentence, It would be helpful to add this reference since this study was about Hong Kong as well. Public transport policy measures for improving elderly mobility”
Response:
We have revised conclusion section. Thank you very much for the very helpful paper suggested! We have cited the paper and also added the findings from the paper (i.e., shortening waiting time and improving seats availability) as the policy recommendation. Thanks!
Round 2
Reviewer 3 Report
I have no more comments.
This manuscript is a resubmission of an earlier submission. The following is a list of the peer review reports and author responses from that submission.
Round 1
Reviewer 1 Report
Revision of the paper: Activity space based on GPS trajectory data: A comparative 2 study of poor and non-poor older adults in Hong Kong
This paper deals with a novelty and interest topic by exploring mobility through the lens of social exclusion.
Major concern: why ageing is always understood as a problem? Why we cannot assume that ageing and living longer is a positive aspect of living conditions and health sector improvement? We should be careful when discussing ageing issues.
I am quite surprised that the authors have not mentioned in the introduction the impact of symbolic and social barriers in the use of physical space.
Also, why did you use “poor” and “non-poor”? How did you define it? Poor in what dimension? Did you believe this is the best term to use? Did it translate the phenomena that you are trying to measure?
Methods: How did you know that the “poor” have fewer social networks? How did you measure?
The paper is crying for a scheme with the relation between variables, maybe the analytical model behind it. In the current version is hard to follow the authors' rationale.
The results are poorly presented. The information provided is very basic and deserves further exploration. A regression model would be important to predict how mobility may be explained by social exclusion or socioeconomic position.
In the discussion section: Did you consider the Hong Kong city planning as an explanation to find no differences in daily activity? Maybe the city provides close services to participants residences. Also, you did not present any information regarding participants residence located in the city and how the city is organized.
This study needs more work before can be considered publishable.
Good luck with your revisions!
Author Response
Dear Reviewer,
Thank you so much for reviewing our paper. Your valuable comments indeed help us improve the paper tremendously and understand the issues/problems from a more comprehensive perspective. Truly, we appreciate your advice. The following are our response. Please find the detailed replies respectively.
Reviewer 1
- “why ageing is always understood as a problem? Why we cannot assume that ageing and living longer is a positive aspect of living conditions and health sector improvement? We should be careful when discussing ageing issues.”
Response:
Thanks for the insight! We have changed the tone about the discussion on ageing by emphasizing both positive aspect and the challenges.
- “I am quite surprised that the authors have not mentioned in the introduction the impact of symbolic and social barriers in the use of physical space.”
Response:
Thanks for the comment! We have added this in the introduction in the revised manuscript. It is very helpful to improve our paper to a higher level. Thanks!
- “why did you use “poor” and “non-poor”? How did you define it? Poor in what dimension? Did you believe this is the best term to use? Did it translate the phenomena that you are trying to measure?”
Response:
Thanks for the comment. To make it more clear and more related to what we would like to measure, we have changed “poor and non-poor older adults” to “lower- and higher-income older adults” and added exploration in Method, i.e., the lower-income older adults refer to those who live depend on Hong Kong government comprehensive social security assistance (CSSA), while higher-income older adults refer to those who are not CSSA receiver.
- “How did you know that the “poor” have fewer social networks? How did you measure?”
Response:
Thanks! This study is a pilot study testing the incorporation of quantitative GPS trajectory data to descriptively compare the two groups’ mobility pattern, and findings of in-depth semi-structured interviews to quantitively understanding the potential reasons behind the difference between two groups. We used the pilot study design to largely test feasibility of recruitment and measurement. Regarding the social-network, we asked a general open question “whom did you meet with in last one week” and let the participants to share about their experience. In the semi-structured design, we tried to make the question simple and let the participants expand the content, so that we can dig more information from their talking. We found that more in depth questioning was required in subsequent studies, because we did not find the detail we needed.
Regarding the different mobility pattern and underlying reasons, three main themes were identified by thematic analysis based on semi-structured interview. The first theme is about social network, i.e., seven cases mentioned that their social circle was small and they didn’t contact with friends frequently in lower-income group, while two cases reported that they visit friends often in higher-income group.
We have revised the manuscript substantially by emphasizing the mixed-design and adding more information about qualitative part in title, methods, and results.
- “The paper is crying for a scheme with the relation between variables, maybe the analytical model behind it. In the current version is hard to follow the authors' rationale.”
Response:
Thanks! To make it more clear, we have revised the manuscript substantially by emphasizing the pilot aims with comparative and mixed-approach design more. We aim to test recruitment processes, and test feasibility of extracting mobility indicators, and test the feasibility of using qualitative interview information to deeper understanding the underlying reasons of the difference of mobility pattern between lower- and higher- income older adults.
Therefore, we were not to collect questionnaire information to conduct regression analysis based on various variables. Instead, we used GPS data to descriptively compare the objective mobility patterns between two groups and applied semi-structured interview to understand the underlying reason.
- “The results are poorly presented. The information provided is very basic and deserves further exploration. A regression model would be important to predict how mobility may be explained by social exclusion or socioeconomic position.”
Response:
Thanks for the comments! We have revised the manuscript substantially by emphasizing the pilot nature of the study, and ways of combining qualitative and quantitative data to understand mobility. In the discussion, we have added future directions including testing a larger representative sample to more carefully understand mobility choices and factors leading to it.
- “In the discussion section: Did you consider the Hong Kong city planning as an explanation to find no differences in daily activity? Maybe the city provides close services to participants residences. Also, you did not present any information regarding participants residence located in the city and how the city is organized.”
Response:
Thanks! We have added the city background information in the introduction and emphasized more about the effect of city context on the findings in discussion, and added the participants’ residence locations information in method and a background map in results.
Reviewer 2 Report
Sample size chosen on what basis? I think the sample size better be increased. Female percentage is high among sample. Elaborate the variables measuring the four characteristics of activity space by figure or present the finding in figure (for example give a diagram of Standard deviational ellipse (SDE) of one respondent as a sample demonstration of activity space calculation and also the Standard distance between daily activity locations of one respondent at least). Also, define the activity space calculation methods used here in the 2.2 Measures section. Result section should include more tables/figures if possible. Overall, it's a good work with new idea covering the unnoticed segment of activity space research. Paper can be accepted with minor revisions.
Author Response
Dear Reviewer,
Thank you so much for reviewing our paper. Your valuable comments indeed help us improve the paper tremendously and understand the issues/problems from a more comprehensive perspective. Truly, we appreciate your advice. The following are our response. Please find the detailed replies respectively.
- “Sample size chosen on what basis? I think the sample size better be increased. Female percentage is high among sample.”
Response:
Thanks! To make it more clear, we have revised the manuscript substantially by emphasizing the comparative and mixed-design more. In this study, our main aim was to descriptively compare the mobility pattern between two groups and understanding the underlying reasons in more-depth based on qualitative interview. Therefore, we were not to collect questionnaire information to conduct regression analysis based on various variables. Instead, we used GPS data to descriptively compare the objective mobility patterns between two groups and applied semi-structured interview to understand the underlying reason. For this objective, we used purposive sampling method to recruit participants. The main criterion of the selection of participants was to cover both lower- and higher-income groups of healthy older adults aged 60 or older who can live independent lives without need for walking assistance.
- “Elaborate the variables measuring the four characteristics of activity space by figure or present the finding in figure (for example give a diagram of Standard deviational ellipse (SDE) of one respondent as a sample demonstration of activity space calculation and also the Standard distance between daily activity locations of one respondent at least).”
Response:
Thanks! We have added figure to illustrate the SDE and Standard distance.
- “define the activity space calculation methods used here in the 2.2 Measures section.”
Response:
Thanks! We have added more details about the calculation methods in section 2.2
- “Result section should include more tables/figures if possible. Overall, it's a good work with new idea covering the unnoticed segment of activity space research. Paper can be accepted with minor revisions.”
Response:
Thanks! We have added Hong Kong background map, participants’ residential areas and the extensity pattern by lower- and higher-income older adults.
Reviewer 3 Report
Thank you for the opportunity to review this manuscript. This study aimed to use GPS tracking data to identify participant’s daily activity space, and small-areal census data to characterize the social environments of activity space, in order to examine and compare the different activity space patterns of poor and non-poor older adults in Hong Kong. While I believe the topic is an interesting one, I have some suggestions and comments which might improve the quality of this paper.
There are a number of grammatical/ syntactic/ proofing issues to attend to when the paper has been reshaped. These are too numerous to list here but would be cleared up through a quality proofing process. Before any further revision of the paper, I suggest the paper needs proofreading by a native English speaker.
In literature review section, most of the key literature are old. It seems to become outdated and less desirable.
- Golledge, R.G. and R.J. Stimson, Spatial Behavior: A Geographic Perspective. New York: Guilford Press, 1997.
- Thornton, P.R., A.M. Williams, and G. Shaw, Revisiting time-space diaries: an exploratory case study of tourist behaviour in Cornwall, England. Environment and Planning A, 1997. 29(10): p. 1847-1867.
- Shoval, N., Tracking technologies and urban analysis. Cities, 2008. 25(1): p. 21-28.
- Shoval, N. and M. Isaacson, Application of tracking technologies to the study of pedestrian spatial behavior. Professional Geographer, 2006. 58(2): p. 172-183.
- Paz-Soldan, V.A., Strengths and Weaknesses of Global Positioning System (GPS) Data-Loggers and Semi-structured Interviews for Capturing Fine-scale Human Mobility: Findings from Iquitos, Peru. 2014.
- Krenn, P.J., et al., Use of Global Positioning Systems to Study Physical Activity and the Environment A Systematic Review. American Journal of Preventive Medicine, 2011. 41(5): p. 508-515.
- Shareck, M., Y. Kestens, and L. Gauvin, Examining the spatial congruence between data obtained with a novel activity location questionnaire, continuous GPS tracking, and prompted recall surveys. International Journal of Health Geographics, 2013. 12.
- Rodriguez, D.A., et al., Identifying walking trips from GPS and accelerometer data in adolescent females. J Phys Act Health, 2012. 9(3): p. 421-431.
I recommended discussing relate works from journal articles in the last five years.
Calabrese et al., 2007 have introduced new developments in the application of advanced remote sensing techniques which can open the way to investigating people’s behaviour in entire urban regions without the need for people to wear GPS devices. The development of mobile phone and applications today has made it possible to investigate people’s behaviour without the need for wearing GPS devices. Why did you still choose this ancient way to investigate the sample of citizens selected in advance to wear the GPS devices?
Is the sample size (n=45) representative enough?
The references were not written in accordance to a typical format of this journal.
Author Response
Dear Reviewer,
Thank you so much for reviewing our paper. Your valuable comments indeed help us improve the paper tremendously and understand the issues/problems from a more comprehensive perspective. Truly, we appreciate your advice. The following are our response. Please find the detailed replies respectively.
Reviewer 3
- “There are a number of grammatical/ syntactic/ proofing issues to attend to when the paper has been reshaped. These are too numerous to list here but would be cleared up through a quality proofing process. Before any further revision of the paper, I suggest the paper needs proofreading by a native English speaker.”
Response:
Thanks! The manuscript has been proofread by a native English speaker.
- “In literature review section, most of the key literature are old (18-25). It seems to become outdated and less desirable. I recommended discussing relate works from journal articles in the last five years.”
Response:
Thanks! We have replaced them with the more recent literature in past five years.
- “Calabrese et al., 2007 have introduced new developments in the application of advanced remote sensing techniques which can open the way to investigating people’s behaviour in entire urban regions without the need for people to wear GPS devices. The development of mobile phone and applications today has made it possible to investigate people’s behaviour without the need for wearing GPS devices. Why did you still choose this ancient way to investigate the sample of citizens selected in advance to wear the GPS devices?”
Response:
Thanks for the information! We have added this in the future direction. In the poverty research filed, GPS is already a very advanced technology for us to conduct the mobility study in daily life, compared with the traditional questionnaire survey and qualitative interview method. In future studies, we will adopt the methods as you suggested. Thank you very much for the information.
- “Is the sample size (n=45) representative enough?”
Response:
Thanks! To make it more clear, we have revised the manuscript substantially by emphasizing the comparative and mixed-design more. In this study, our main aim was to descriptively compare the mobility pattern between two groups based on GPS data and understanding the underlying reasons in more-depth based on qualitative interview. Therefore, we were not to collect questionnaire information to conduct regression analysis based on various variables.
- “The references were not written in accordance to a typical format of this journal.”
Response:
Thanks! We have revised the reference accordingly.
Reviewer 4 Report
1. Introduction
Lines 96-105: it would be beneficial for readers to have a dedicated paragraph to explain: 1) research gap summary, 2) research objective(s), and 3) contribution of this study at the end of intro.
2. Materials and Methods: It would be helpful to have a paragraph by explaining overall process/structure of the methodology adopted/used in this research before 2.1 section.
2.1 Participants: what are the age range of older adults? Table 1 shows age over 60 as an older adult. Typical older adults are persons above age 65?
Author Response
Dear Reviewer,
Thank you so much for reviewing our paper. Your valuable comments indeed help us improve the paper tremendously and understand the issues/problems from a more comprehensive perspective. Truly, we appreciate your advice. The following are our response. Please find the detailed replies respectively.
- “Lines 96-105: it would be beneficial for readers to have a dedicated paragraph to explain: 1) research gap summary, 2) research objective(s), and 3) contribution of this study at the end of intro.”
Response:
Thanks! We have restructured the introduction and added a dedicated paragraph to summarize the research gaps, research objectives, and potential contribution of this study, at the end of introduction in the revised manuscript.
- “It would be helpful to have a paragraph by explaining overall process/structure of the methodology adopted/used in this research before 2.1 section.”
Response:
We have deleted the duplicated part, and repackaged this section mainly focusing on the research objectives and the conceptual framework of the current study. Moreover, to make the subtitle in consistent with the content in this section, we have changed it
- “what are the age range of older adults? Table 1 shows age over 60 as an older adult. Typical older adults are persons above age 65?”
Response:
Yes, typical older adults are persons aged 65 or above. However, due to the difficulty in recruiting the older adults as participants in Hong Kong, we have expanded our criteria to those who aged 60 or above, we have added this in the revised manuscript.